# HIV treatment response among female sex workers participating in a treatment as prevention demonstration project in Cotonou, Benin

Mamadou Diallo[1,2], Luc Béhanzin[2,3,4], Fernand A. Guédou[2,3], Nassirou Geraldo[3], Ella Goma-Matsétsé[3], Dramane Kania[5], René Kpèmahouton Kêkê[6], Moussa Bachabi[6], Dissou Affolabi[7,8], Souleymane Diabaté[1,2,9], Flore Gangbo[6,7,8], Marcel Djimon Zannou[7,8], Michel Alary[1,2,10]*

1 Département de médecine sociale et préventive, Université Laval, Québec, 2 Axe Santé des populations et pratiques optimales en santé, Centre de recherche du CHU de Québec–Université Laval, Québec, Canada, 3 Dispensaire IST, Centre de santé communal de Cotonou 1, Cotonou, Bénin, 4 Ecole Nationale de Formation des Techniciens Supérieurs en Santé Publique et en Surveillance Épidémiologique, Université de Parakou, Bénin, 5 National Reference Laboratory of Viral Hemorrhagic Fever Centre MURAZ, Bobo-Dioulasso, Burkina Faso, 6 Programme Santé de Lutte contre le Sida (PSLS), Cotonou, Bénin, 7 Centre national hospitalier universitaire HMK de Cotonou, Bénin, 8 Faculté des sciences de la santé, Université d'Abomey-Calavi, Cotonou, Bénin, 9 Université Alassane Ouattara, Bouaké, Côte d'Ivoire, 10 Institut national de santé publique du Québec, Québec, Canada

* malary@uresp.ulaval.ca

**Data Availability Statement:** All relevant data are publicly available in the manuscript and its Supporting Information file.

## Abstract

### Objectives

Female sex workers (FSWs) play a key role in HIV transmission in West Africa, while they have limited access to antiretroviral therapy (ART). In line with UNAIDS recommendations extending ART to all HIV-infected individuals, we conducted this demonstration project on immediate treatment as prevention (TasP) among FSWs in Cotonou, Benin. We report data on treatment response and its relation to adherence, as well as on ART-resistant genotypes.

### Methods

Complete follow-up varied between 12 and 24 months. At each three-monthly visit, a questionnaire was administered, clinical examinations were carried out and blood samples collected. Adherence to treatment was estimated by self-report. Viral RNA was genotyped at baseline and final visits for drug resistance. Generalized estimating equations for repeated measures with a log-binomial link were used to analyze time trends and the association between adherence and virological response to treatment.

### Results

One-hundred-seven HIV-positive and ART-naive FSWs were enrolled; 59.8% remained in the cohort till study completion and 62.6% had a final visit. Viral load<1000 (below

**Funding:** This study was funded by the Bill and Melinda Gates Foundation (grant # OPP1098973). Complementary funding was provided by the Canadian Institutes of Health Research (grants #ROH-115205 and # FDN-143218). SD Bioline HIV/Syphilis duo test was provided free of charge by Standard Diagnostics.

**Competing interests:** SD Bioline HIV/Syphilis duo test was provided free of charge by Standard Diagnostics. This does not alter our adherence to PLOS ONE policies on sharing data and materials.

quantification limit [<50]) was attained in 73.1% (64.6%) of participants at month-6, 84.8% (71.2%) at month-12, and 80.9% (65.1%) at the final visit. The proportion of women with suppressed (below quantification limit) viral load increased with increasing self-reported adherence (p = 0.06 (0.003), tests for trend). The proportion of participants with CD4$\leq$500 also decreased drastically throughout follow-up (p < .0001). Twelve participants exhibited ART-resistant genotypes at baseline, but only two at their final visit.

## Conclusion

Our findings indicate that TasP is widely accepted among FSWs in Cotonou and could be implemented with relative success. However, due to mobility in this population, follow-up was sub-optimal, suggesting that large geographical coverage of FSW-friendly clinics is needed for sustained treatment implementation. We also fell short of the UNAIDS objective of 90% viral suppression among treated patients, underlining the need for better adherence support programs.

## Introduction

Western and central Africa is the second most affected region by the HIV epidemic with an estimated 5 million people living with HIV (PLHIV) at the end of 2018 (representing 13.2% of all HIV-infected people in the world). Yet, HIV testing and antiretroviral therapy (ART) coverage in this region are among the lowest worldwide [1]. Overall, the UNAIDS 2020 targets remain far from being attained with only 42% of the PLHIV knowing their status, 35% (83% of those knowing their HIV status) on ART, and only 25% (71.4% of those on ART) having suppressed their viral load [2]. Many countries of this region are characterized by an HIV epidemic where specific behaviours relevant to key populations are the main epidemic drivers [3–5]. Female sex workers (FSWs) and their clients, alongside less visible groups involved in part-time transactional sex, including bar workers and mobile fruit sellers [6], play a central role in the HIV transmission dynamics [7, 8]. Several reports in the region suggest that FSWs contributed to between 32–84% of HIV prevalent cases among men [9–11] and to 18% in the general population of women (12), providing evidence that prioritizing this high-risk group with specific preventive interventions could reduce HIV incidence and prevalence in both prioritized groups and the general population at lower risk [12–14]. However, despite such contribution, ART coverage among FSWs remains low in the region [15, 16] For these reasons, we considered FSWs as a priority population for the implementation of new biomedical preventive interventions such as immediate HIV treatment as prevention (TasP) and pre-exposure prophylaxis (PrEP) [17–19]. We conducted a TasP/PrEP demonstration project to assess the acceptability, feasibility, and utility to integrate TasP and PrEP among current prevention methods in FSWs in Cotonou. The present paper reports the main findings of the TasP component of this demonstration project [20] including virological and immunological response to treatment, and resistance to antiretroviral drugs.

## Material and methods

### Study design

Recruitment was conducted in pre-specified areas of Greater Cotonou (Cotonou and close suburbs) in order to approximately meet target sample sizes of 250 for the PrEP arm, and 100

for the TasP arm. After a run-in phase for community preparedness and the development of a specific education program on adherence, FSWs from the catchment areas were invited to attend the "Dispensaire IST", a clinic dedicated to FSWs for HIV and sexually transmitted infections (STIs) care in Cotonou, for HIV testing. HIV-positive FSWs not known to be on ART were then invited to participate in the TasP component of the study and received a first-line ART regimen as per the Benin guidelines. Recruitment took from October 2014 to December 2015, followed by an additional one-year follow-up till December 2016, for a total follow-up varying between 12 and 24 months, depending on when a given participant was recruited in the study. A short-term follow-up visit to assess adherence and drug adverse events was scheduled at day 14 with subsequent visits being scheduled quarterly. At baseline and during follow-up visits, clinical examinations were carried out, vaginal and blood samples collected for STIs testing, CD4 count, and viral load quantification. Adherence was assessed during follow-up visits by self-report through a questionnaire. Participants who came back to the clinic for withdrawal were invited to undergo all the procedures of a final visit, while those who stopped participating without informing the clinic, were located by field workers and reasons for dropping out documented.

## Study population and eligibility criteria

The study population consisted of professional FSWs (e.g. whose main revenue comes from sex work) 18 years or older, practicing in the pre-defined catchment areas. Those eligible for TasP had to be HIV-1 positive at screening, re-confirmed on a second test at the recruitment visit, and being treatment naïve. The only exclusion criterion was HIV-2 or dual HIV-1/2 infection.

## Data collection

**Demographic, behavioural and sex work characteristics.** Several socio-demographic and sex work characteristics including nationality, age, marital status, education, parity, activities beside sex work, duration in sex work and monthly income were assessed through a questionnaire at baseline. Sexual behaviour including number of clients in the last two weeks, having a regular sexual partner and condom use with both clients and regular partners were assessed through the questionnaire at baseline and during follow-up. In order to assess adherence, at each visit, questions were asked on the number of pills missed in the last month.

**Laboratory procedures at the different visits.** At screening, all subjects were tested for HIV and syphilis using the SD Bioline HIV/syphilis Duo test (Standard Diagnostics, Yongin, South Korea, which is a treponemal test with confirmation of HIV-positive tests using the Immunoflow HIV-1/2 test (Core Diagnostics, Birmingham, UK). They were also tested for anti-HBc and anti-HBs antibodies and HBs antigen (HBsAg) using the Monolisa-Biorad tests of Biorad (Marnes-la-Coquette, France). FSWs initially testing positive using the treponemal rapid test were then tested with the Rapid Plasma Reagin (RPR) non-treponemal (also provided by Standard Diagnostics). Women testing positive on both the rapid and RPR tests were considered as cases of active syphilis. Subjects negative to hepatitis B antibodies and antigen were vaccinated against this infection. HIV seropositivity was confirmed on a separate blood finger-prick sample at recruitment using the same tests as above. At recruitment and during follow-up visits, vaginal swabs were collected for direct microscopic examination for yeasts and *Trichomonas vaginalis*; and abnormal vaginal flora using the Nugent score for the diagnosis of bacterial vaginosis. Participants also underwent testing for *Neisseria gonorrhoeae* and *Chlamydia trachomatis* on cervical swabs at recruitment and every six months using the BD Probetec NG/CT assay (Becton Dickenson, Cockeysville, MD, USA). Additional tests

including glycaemia, blood creatinine serum levels (renal function), and alanine aminotransferase serum levels (liver function), as well as Chest-X ray prior to ART initiation were conducted, and complete blood count carried out using standard assays. Blood samples were collected and stored at screening and final visits for future drug resistance genotyping. In the context of the present study, CD4 count was monitored at screening and every three months using a CyFlow Counter (Partec, Germany), whereas viral load was quantified at the screening visit and then every six months using the NucliSens EasyQ equipment (BioMérieux Laboratories, Marcy l'Etoile, France). The technique allows the quantification of a minimum of 50 copies/ml of HIV-1 RNA in 0.5 ml plasma [21]. For both tests (CD4+ T-cell counting and HIV-1 viral load), positive and negative controls were used in each run. All plasma samples collected at screening and those at final visits with a viral load > 1000 copies per mL were tested for drug resistance. Samples were shipped on dry ice to the Centre Muraz, Bobo Dioulasso, Burkina Faso, where genotyping was carried out using a method developed by the *Agence nationale de recherche sur le Sida (ANRS)*, the French agency for AIDS research described in details on the ANRS website [22]. The interpretation of resistance mutations used the latest updates of the three main algorithms currently available: one from ANRS [23], one from Stanford University [24, 25] and one from the Rega Institute for Medical Research in Leuven, Belgium [26].

**Treatment regimens.** The antiretroviral regimen consisted of Tenolam E® (Hetero Healthcare Ltd., Hyderabad, India) tablets with two nucleotides reverse-transcriptase inhibitors (NRTIs)

(Tenofovir, Lamivudine) and one Non-nucleoside reverse-transcriptase inhibitors (NNRTIs) (Efavirenz) as active ingredients. In case of suspicion of resistance (repeated viral load above 1000 copies/ml; decrease, or lack of increase in CD4 count), a second line treatment was suggested with Lopinavir /Ritonavir (+ lamivudine and tenofovir). For those who desired pregnancy, Tenofovir, Lamivudine and Nevirapine were suggested. Their ART regimen was provided to the participants for a 30-day period. Outside periods of scheduled follow-up visits, they were contacted by field workers every month and, at that time, they were given the choice to come to the clinic just to pick up another monthly supply or to have it delivered at home. All participants diagnosed with STIs were treated immediately according to the Benin national guidelines.

**Definition of outcome variables.** The primary outcome of interest was the virological response defined as (i) suppressed (<1000 copies/mL) as suggested by the WHO 2013 Consolidated guidelines on the use of antiretroviral drugs for treating and preventing HIV infection [27], or (ii) below quantification limit of the assay used during the study. In this study, since we used NucliSENS EasyQ® HIV-1 v2.0 (Semi-Automated), and 0.5ml plasma for viral load quantification, the "quantification limit" was 50 copies/ml. Therefore, all samples in which we could not amplify the viral load were considered as having a viral load below quantification limit (below 50 copies/mL). In relation to this first outcome, we analyzed the evolution of viral load throughout follow-up as well as the association between self-reported adherence and viral load. The second outcome was changes in CD4 count throughout follow-up, considering a CD4 count fall to (or below) the baseline value or persistent CD4 levels below 100 cells/mm$^3$ as indicative of immunologic failure [27]. The last outcome was the development of drug resistance during the study, but we also report on the presence of primary drug resistance at baseline.

## Statistical analysis

All data were analyzed using the SAS 9.4 statistical software (SAS Institute, Cary, NC, USA). First, we used descriptive statistics to estimate means and medians for continuous variables,

while categorical variables were described using proportions. In particular for immunological and virological parameters, means and medians were estimated using the actual CD4 count and the log of viral load values (the viral load value was set at 50 when below quantification limit) to obtain geometric means for the latter, while proportions were obtained by categorizing the CD4 count into ≤500 cell/μl vs >500 cell/μl; and the viral load into suppressed (<1000 copies/ml) vs unsuppressed, and below quantification limit (<50 copies/ml) vs above quantification limit. We used generalized estimating equations (GEE) models for repeated measures with a log-binomial link and type3 WALD statistical tests to assess trends in CD4 and viral load throughout follow-up, as well as the association between viral load and self-reported adherence while controlling for time duration since ART initiation and other potential confounders, such as age, country of origin and several sexual behaviour variables. Self-reported adherence was categorized into less than 3 pills missed in the last month, corresponding to high adherence (adherence ≥90%), 4–7 pills missed corresponding to medium adherence (adherence between 75–89%), 8–15 pills missed corresponding to low adherence (adherence between 50–74%), and >15 pills missed corresponding to non-adherence (adherence below 50%).

## Ethical considerations

After detailed explanation of the study, women provided informed written consent for the screening visit and then again at recruitment for the whole follow-up period. The participants were free to withdraw from the study at any time without any prejudice. The study was approved by the National Committee of Ethics in Health Research of Benin and by the Institutional Review Board of the CHU de Québec–Université Laval, Québec, Canada.

This study is registered with ClinicalTrials.gov, number NCT02237027.

# Results

## Recruitment and follow-up

Out of the 111 participants tested HIV-positive at screening, 5 declined to participate and one was infected with both HIV-1 and HIV-2, thus excluded from the study and referred for standard treatment. During follow-up, two participants from the PrEP arm seroconverted and were included in the TasP program, summing up to a total of 107 participants recruited for TasP. Participants were then followed for a mean duration of 13.2 ± (Standard Deviation 7.7) months. During follow-up, seven participants withdrew for not being interested to the study anymore (6.5%), 15 went back to their country of origin (14.0%), 19 went to others cities to practice sex work (17.8%), one died and one got married and left sex work; leaving a final number of 64 participants who effectively completed the study. This makes up a strict retention rate of 59.8% and global retention rate of 62.6% since three participants who actively withdrew underwent all the procedures of final visits.

## Baseline characteristics of participants

Baseline sociodemographic characteristics of participants are shown in Table 1. Almost half of them were foreigners; median age was 35 years [Interquartile range (IQR): 30–42], and about 40% did not have any formal education. About 70% were either divorced or widowed, and over 80% had at least one child. The median duration of sex work was 2 years (IQR: 0.46–3), with 32.7% being on sex work for less than one year, and 13.0% for more than five years. The median number of sexual clients in the last two weeks was 12 [IQR: 4–30] with 10.3% not having any sexual client during that period; and condom use at last sexual act was reported by

**Table 1. Baseline characteristics of FSWs participating in the TasP demonstration study in Cotonou, Benin.**

| Variable | Total (n) | Total (%) |
|---|---|---|
| **Country of origin** | **107** | **100%** |
| Benin | 56 | 52.3% |
| Other countries | 51 | 47.7% |
| **Age (in years)** | **107** | **100%** |
| 18–24 | 13 | 12.1% |
| 25–34 | 35 | 32.7% |
| 35–44 | 40 | 37.3% |
| ≥45 | 19 | 17.7% |
| Mean age ± SD[a] | 35.5 ±8.8 | |
| Median age [IQR[b]] | 35 [30–42] | |
| **Marital status** | **107** | **100%** |
| Married | 2 | 1.8% |
| Divorced or Widowed | 76 | 70.9% |
| Single | 29 | 27.1% |
| **Parity** | **105** | **98.1%** |
| 0 child | 9 | 8.6% |
| 1 child | 27 | 25.7% |
| 2–3 children | 37 | 35.2% |
| 4–5 children | 24 | 22.8% |
| >5 children | 8 | 7.6% |
| **Variables related to sex work and sexual behavior** | | |
| **Duration of sex work (years)** | **105** | **98.1%** |
| <1 | 35 | 33.3% |
| 1–5 | 56 | 53.3% |
| >5 | 14 | 13.3% |
| Median duration of sex work, [IQR[b]] | 2 years [0.46–3.0] | |
| **Number of sexual clients during the last 14 days** | **97** | **90.6%** |
| 0 | 11 | 11.3% |
| 1–4 | 13 | 13.4% |
| 5–9 | 15 | 15.4% |
| 10–19 | 19 | 19.6% |
| 20–49 | 28 | 28.8% |
| ≥ 50 | 11 | 11.3% |
| Median number of sexual clients, [IQR[b]] | 12 [5–30] | |
| **Condom use with last client** | **105** | **98.1%** |
| Yes | 102 | 97.1% |
| **Had a regular sexual partner in the last 12 months** | **105** | **98.1%** |
| Yes | 58 | 55.24% |
| **Condom use with regular partner at last sex** | **57** | **53.2%** |
| No | 48 | 84.2% |
| **Monthly income in FCFA[c]** | **103** | **96.2%** |
| ≤100.000 | 36 | 34.9% |
| 101.000 à 200.000 | 41 | 39.8% |
| >200.000 | 26 | 25.2% |
| Mean Income ±SD[a] | 170,000 ±100.00 | |
| Median Income [IQR[b]] | 150,000 [90,000.00–210,000.00] | |
| **Sexually transmitted infections (STIs[d])** | | |

*(Continued)*

**Table 1.** (Continued)

| Variable | Total (n) | Total (%) |
|---|---|---|
| *N. gonorrhoeae* | 104 | 97.2% |
| positive | 6 | 5.7% |
| *C. trachomatis* | 104 | 97.2% |
| positive | 3 | 2.8% |
| Trichomonas *vaginalis* | 104 | 97.2% |
| positive | 1 | 0.9% |
| **Vaginal Candidiasis** | 104 | 97.2% |
| positive | 8 | 7.2% |
| **Bacterial vaginosis** | 104 | 97.2% |
| Normal Flora (Nugent score 0–3) | 10 | 9.6% |
| Intermediate Flora (Nugent score 4–6) | 30 | 28.8% |
| Bacterial vaginosis (Nugent score 7–10) | 64 | 61.5% |
| **Serology** | | |
| **Active Syphilis** | **107** | **100%** |
| Positive | 0.0 | 0.0% |
| **Anti_HBs** | **107** | **100%** |
| Positive | 48 | 46.6% |
| **Anti_HBc** | **107** | **100%** |
| Positive | 86 | 83.5% |
| **Ag HBs** | **107** | **100%** |
| Positive | 5 | 4.6% |

[a]**SD**: Standard Deviation

[b]**IQR**: Interquartile range

[c]**F CFA**: Franc CFA (name of Beninese currency); US$1 is equivalent to about 500 F CFA

[d]**STIs**: Sexually transmitted Infections

**Note**: Due to missing values, the total number (**N**) for each variable may be different from 107. However, the few missing values could not significantly affect the results.

97.1% of participants. The median monthly income was 150,000.00F CFA [IQR: 90,000.00–210,000.00], about USD $270 (IQR: 165–380), with 34.9% earning less than ≤100.000 FCFA per month, about USD $200. Nearly 7.7% of participants were infected by either *N. gonorrhoeae* or *C. trachomatis*; 7.2% were positive for vaginal candidiasis, 1% for trichomoniasis; and two thirds had bacterial vaginosis (Nugent Score 7–10). Only 15.5% had WHO clinical stage II with either CD4 count below 200 cell/µl, presence of emaciation, dermatosis, persistent cold, prolonged fever, oral thrush, or cutaneous mucosal lesions. No participant was diagnosed with active syphilis, 13.1% were negative to both anti-HBc and anti-HBs and about 5% had active hepatitis B (see Table 1). All participants started ART with Tenolam-E®.

## CD4 count

The mean CD4 count increased significantly throughout follow-up, from 526.5 (Standard deviation: 329.2) at baseline to 718.1 (standard deviation: 385.8) at month 24 (p trend < .0001) while the proportion of participants with CD4<500 decreased significantly from 53.2% to 29.4% during the same period (p trend = 0.008) (Fig 1). In addition, among the 64 participants who had regular final visits (either a premature final visit or a final visit at the end of the study), the proportion with CD4 count <500 at that visit was significantly lower (27.9%)

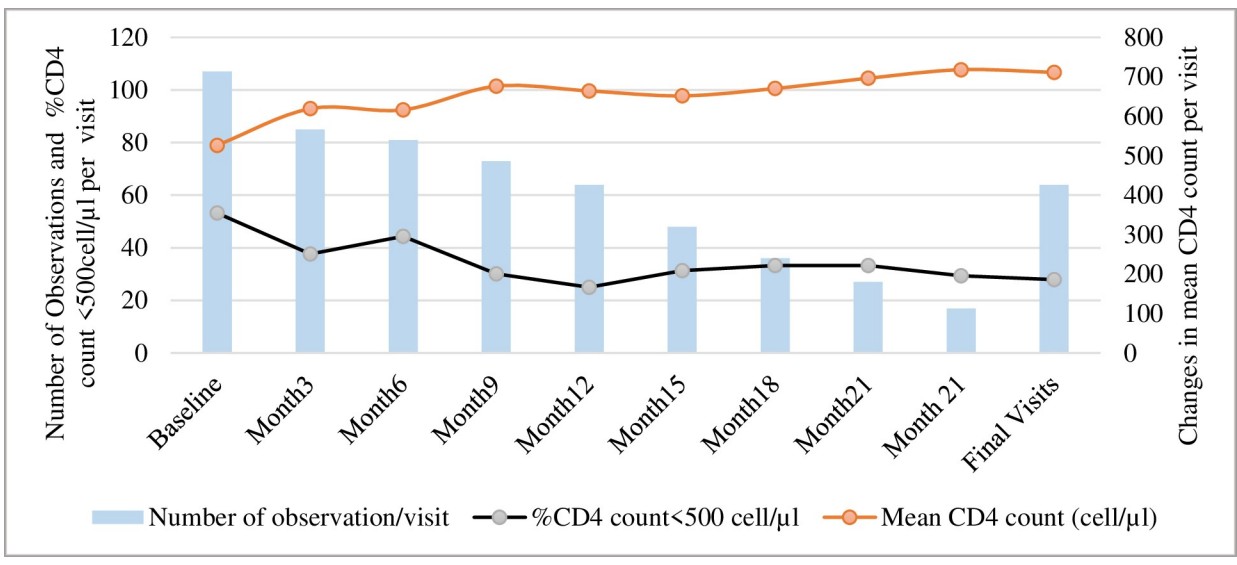

**Fig 1. Time trends in mean CD4 and proportions of CD4<500 among HIV-infected FSWs on ART, Benin. N:** Number of observations per visit **%CD4 count**<**500 cell/µl:** Proportion of participants with CD4<500 cell/µl during follow up **Mean CD4 count:** Changes in mean CD4 count during follow up**Final visits:** Included 63 participants who had final visits from Month 3 onwards and one early final that occurred at Month 3. Subjects with final visits at Day 14 (one at the end of the study for a seroconverting participant from the PrEP component of the study and two early final visits because of active withdrawal) were excluded from this analysis.

compared to baseline (p < .0001) while the mean CD4 count was significantly higher (711.3 vs. 526.5, p < .0001). However, despite such trends in CD4 recovery, three participants experienced immunological failure with persistent CD4 count below 100 cells/µl even after one year of treatment. These participants started treatment with low CD4 counts at baseline, respectively 19, 28, and 52 cells/µl; had adherence level below 50% or sometimes missing; and had all maintained viral load above 1000 copies/ml during follow-up.

## Viral load

Out of the 107 samples collected at baseline, 19 (17.8%) had suppressed viral load, and 3 (2.8%) had viral load below quantification limit. At month 6, 73.2% (64.6%) of the 82 participants still followed up, had suppressed (<1000 copies/mL) and below quantification limit (<50 copies/mL) viral load. These percentages increased respectively to 84.8% and 71.2% among the 66 participants remaining in the study after one year (88.2% and 82.3% at month 24). The geometric mean of viral load decreased significantly throughout follow-up, from 12372.0 [95% Confidence Interval (95%CI): 7575.2–202017.4) to 95.1 (95%CI: 31.0–291.0) at month 24 (p trend < .0001), while there was significant increase in the proportions of both suppressed (<1000 copies/mL) (p trend < .0001), and viral load below quantification limit (<50 copies/mL) (p trend < .0001). The geometric mean of viral load at final visits (for both those with a premature final visit and those with a regular final visit) was also significantly lower (144.4; 95%CI: 79.8–261.4) compared to baseline (p < .0001) and over 81.0% (65.1%) of participants with regular final visits had suppressed (below quantification limit) viral load (p < .0001) (Fig 2). However, six participants, who initially suppressed their viral load at month 6, experienced viral rebound at month 12. During follow-up, eight participants experienced virological failure with sustained viral load above 1000 copies/mL for more than two consecutive visits.

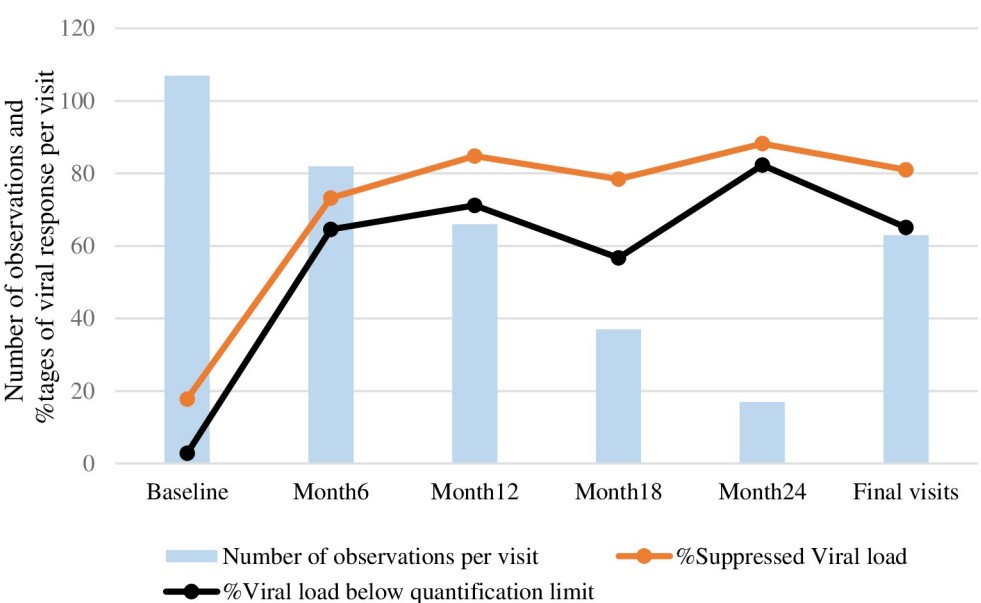

**Fig 2. Time trends in suppressed and below quantification limit viral load among HIV-infected FSWs on ART, Benin. N:** Number of observations per visit **% Suppressed viral load**: Proportion of participants with viral load <1000 copies/ml **% Below quantification limit**: Proportion of participants with viral load <50 copies/ml **Final Visits**: Included 63 participants with regular final visits from Month 6 onwards; excluded all final visits that occurred prior to M6.

## Adherence

Among the 217 follow-up visits for which viral load testing was carried out at month 3 or later, data on self-reported adherence was available for 207 (95.4%). Table 2 shows the association between viral load and self-reported adherence using all observations while controlling for study visit rank (the other potential confounders were dropped from the final model because they did not change by more than 10% the prevalence ratios between the different adherence levels and the prevalence of suppresses or below quantification limit viral load. The proportion of visits where viral load was suppressed or below quantification limit increased with increasing adherence (p = 0.06 and 0.003, respectively) (Table 2). However, this increase was mostly observed when adherence levels went from < 50% to 50–74%, with the prevalence ratios remaining almost identical for the two higher adherence categories.

## HIV strains and drug resistance (details in Table 3)

Genotyping for pre-ART drug resistance was attempted on all HIV-positive samples at the screening visit. Out of the 111 samples collected during screening visits and sent for sequencing, 98 (88.3%) were successfully genotyped. Of these, 58.9% were infected with HIV-1 Circulating Recombinant Forms (CRFs) CRF02_AG/CRF02_AG, 7.4% with G/G subtype, 4.2% with CRF01_AE/CRF02_AG subtype, 1.0% with D/D subtype, 4.2% with subtype G/CRF02_AG, and the remaining 24.2% with minor CRFs including CRF02_AG/CRF18_cpx, U/CRF02_AG, CRF06_cpx/CRF02_AG, and CRF11-cpx/CRF02_AG subtypes. Twelve exhibited resistant genotypes making a prevalence of 10.8% (12/111). Among these twelve, four (3.6%) showed resistance to only NRTIs, eleven (10%) to NNRTIs, and one to protease inhibitors (PIs) (Table 3). Mutations associated with resistance to NRTIs included M41L, D67N, T69D, 70R, M184V, T215F, K219Q, T215TS M184I; those associated to NNRTIs included K103N, V179E, Y181C, Y181YF, Y181F, G190A, A98G, Y188L, P225H, Y181S, and those

**Table 2. Prevalence ratio of below quantification limit and suppressed viral load according to self-reported adherence levels among FSWs participating in the TasP demonstration study in Benin.**

| Adherence levels (self-reported) | Prevalences | PR[a] | 95%CI[b] | | p value | p trend[c] |
|---|---|---|---|---|---|---|
| **Suppressed viral load** | | | | | | |
| ≥90% (Pill missed ≤3) | 129/155 (83.2%) | 1.4 | 1.0 | 2.0 | 0.04 | 0,06 |
| 75–89% (Pill missed = 4–7) | 16/19 (84.2%) | 1.4 | 0.9 | 2.2 | 0.10 | |
| 50–74% (Pill missed 8–15) | 9/11 (81.8%) | 1.4 | 0.9 | 2.2 | 0.12 | |
| <50% (Pill missed >15) | 13/22 (59.1%) | 1 | - | - | (Reference) | |
| missing[e] | 5/10 (50.0%) | – | – | – | | |
| **Below quantification limit viral load** | | | | | | |
| ≥90% (Pill missed ≤3) | 114/155 (73.5%) | 3.2 | 1.5 | 6.8 | 0.002 | 0,003 |
| 75–89% (Pill missed = 4–7) | 14/19 (73.7%) | 3.3 | 1.5 | 7.0 | 0.003 | |
| 50–74% (Pill missed 8–15) | 8/11 (72.7%) | 3.3 | 1.5 | 7.1 | 0.003 | |
| <50% (Pill missed >15) | 5/22 (22.7%) | 1 | - | - | (Reference) | |
| missing[e] | 4/10 (40.0%) | – | – | – | | |

[a]**PR**: Prevalence ratio

[b]**95%CI**: 95% Confidence Interval

[c]*p trend*: trend in the prevalence ratio

[e]**Missing**: No data available for self-reported adherence for these subjects

associated to PIs included the major resistance mutation L90M, and other minor resistance mutations L33F, K20I and L10V. All the 12 participants with baseline drug resistance had viral load above 1000 copies/ml, and only one was infected with CRF11-cpx/CRF02_AG strain, the other being infected with either CRF02_AG/CRF02_AG or G/CRF02_AG.

At the end of the study, 12 samples with viral load above 1000 copies/ml were sent for genotyping. Out of them, 10 were successfully genotyped while two were not, despite many attempts. Among those successfully genotyped, two (20.0%) exhibited major mutations associated with resistance to NNRTI or NRTI. Mutations associated with resistance to NRTIs included K70E, M184V, and D67G; those associated to NNRTIs included K103N, V179E, Y188L, V106M, G190A; V82L, and G73GA. These two participants experienced treatment failure and had self-reported adherence levels either missing or very low, varying between 50 and <90%. Among the 12 participants with resistance mutations at baseline, the four who had final visits had all suppressed their viral load.

## Discussion

Our study, one of the first in the West and Central Africa region, revealed wide acceptance of early HIV treatment as prevention (TasP) among FSWs in Cotonou, Benin. This strategy, coupled with adherence to treatment, resulted in restoration of CD4 count, suppression at below quantification limit of viral load, and low emergence of drug resistance. Of note, the World Health Organisation (WHO), in the 2016 WHO Consolidated guidelines on the use of antiretroviral drugs for treating and preventing HIV infection, defines a viral load <1000 copies/mL as treatment success [28]. The term "undetectable" viral load was considered as a viral load below the quantification limit of the assay used in the study. FSWs in this region are disproportionally affected by the HIV infection, play a key role in its spread to the general population [29] but have limited access to antiretroviral therapy [15, 16]. It is in this context that, in line with the recent UNAIDS recommendations extending antiretroviral therapy to all HIV-infected individuals regardless their CD4 count levels [30], we conducted the present

**Table 3. Summary of pre-ART drug resistance mutations among FSWs participating in the TasP demonstration study in Benin.**

| Baseline | | | Mutations | | | Resistance levels | | | HIV Subtypes |
|---|---|---|---|---|---|---|---|---|---|
| Sample ID | CD4 count (cell/µl) | Viral load (copies/ml) | NRTI[a] | NNRTI[b] | PIs[c] | NRTI | NNRTI | PIs | RT / PR regions[d] |
| CD009 | 321 | 11220 | M41L, D67N, T69D, 70R, M184V, T215F, K219Q | K103N, V179E | | Intermediate to High | High | | G/CRF02_AG |
| CD011 | 700 | 6309 | | K103N, Y181C | | | Intermediate to High | | CRF02_AG/ CRF02_AG |
| CD068 | 305 | 2800 | T215TS | Y181YF, G190A | | low | Low to High | | G/CRF02_AG |
| CD083 | 506 | 2786 | T215S | A98G V179E Y188L | | Low | High | | CRF02_AG/ CRF02_AG |
| CD140 | 395 | 140000 | | Y181YF | | | Intermediate to High | | CRF02_AG/ CRF02_AG |
| CD191 | 90 | 36 000 | | Y181YF | | | Low to High | | CRF02_AG/ CRF02_AG |
| CD244 | 720 | 1900 | | K103N, P225H | | | High | | CRF02_AG/ CRF02_AG |
| CD330 | 78 | 100968 | | Y181S | | | Low to High | | CRF02_AG/ CRF02_AG |
| CD333 | 686 | 3466 | M184I | Y181F | | Low to high | Low to High | | CRF11-cpx/ CRF02_AG |
| CD343 | 448 | 2044 | | K103N | | | High | | CRF02_AG/ CRF02_AG |
| CD371 | 1365 | 1000 | | Y181YF | | | Low to High | | CRF02_AG/ CRF02_AG |
| CD314 | 590 | 12657 | | | L90M, K20I, L33F | | | High | CRF02_AG/ CRF02_AG |

[a]**NRTI:** Nucleoside Reverse Transcriptase Inhibitors

[b]**NNTI:** NoNucleoside Reverse Transcriptase Inhibitors

[c]**PIs**: Protease Inhibitors

[d]**RT / PR regions**: Sequencing was conducted at the Protease (PR) and the reverse transcriptase (RT) regions

demonstration project among FSWs in Cotonou Benin. Globally, if TasP was accepted by the majority of HIV-infected FSWs, its feasibility was highly affected by mobility in this group of population. For instance, despite 96.4% enrolled (107/111) at the beginning of the follow-up, about 40.2% dropped out between recruitment and the end of the study. Reasons for dropping out were not necessarily related to TasP intervention itself, rather to seeking clients in other cities or countries. Considering the diversity of origin of FSWs included in this study, this could be predictable since almost half of them were not Beninese nationals. Furthermore, even when they were Beninese, they had frequent trips to other cities within the country (or even moved from Cotonou to such cities) to practice sex work, trips during which their access to treatment may be limited by the absence of se-worker dedicated clinics.

Other similar studies have also shown that loss to follow up is very common in FSWs and a recent TasP demonstration project carried out in South Africa among FSWs also found similar results with one third of the TasP participants not completing their follow-up visits [31]. Indeed, mobility in FSWs is mostly associated with work location (number of clients), being HIV-positive, earning less per client, and experiencing violence among other reasons [32–34], which are not necessarily associated with treatment program. Therefore, the rate of dropout observed in this study should not be interpreted as lack of acceptability or feasibility itself since

more than 90% of FSWs who were offered TasP accepted it, and reasons for dropping out were related to mobility, not to drug uptake. However, this suggests that future studies or interventions should consider mobility as a serious barrier to the intervention. Indeed, only few cities in West Africa have sex-worker dedicated clinics that provide ART services, such as Dispensaire IST in Cotonou. In other clinical settings, FSWs could face individual, social, or structural barriers, including anxiety, stigmatization, lack of social support, violence, or discrimination even from medical staff, that could prevent them from being fully adherent [35, 36]. For the participants who were relatively stable in Cotonou during the study period, such barriers seemed to be unlikely at Dispensaire IST as most of our participants achieved relatively good virological response with about 4 in 5 participants having suppressed, and 3 in 4 having below quantification limit viral load at month 12. Similar studies investigating combination HIV prevention and care package along with systematic reviews have also shown that the fraction of FSWs on ART who reached a below quantification limit viral load ranged between 40%–82% across varying time periods and definitions of viral suppression ($\leq$50 copies/ml, $\leq$100 copies/ml, $\leq$180 copies/ml, $\leq$300 copies/ml)[37–39]. A strong association and a positive trend were found between self-reported adherence and viral suppression. However, this association was weaker for suppressed compared to below quantification limit viral load. This could be explained by the fact that viral suppression is a transition step, while below quantification limit is permanent stage when patients keep taking their drugs. Following ART initiation, most of patients suppress their viral load, and when the medication is taken on a regular basis, the viral load reaches below quantification limit levels where it is maintained as long as patients keep taking their medication. Viral suppression is mainly associated with control of transmission [40, 41], while below quantification limit levels, in addition to reducing transmission, are associated with well-being, meaning better health [42] and a stronger immune system [43]. On the other hand, viral suppression, coupled with both reduced opportunistic infections and good recovery of CD4 levels, could provide a sensation of absolute well-being and health trust that in return pushes such participants to be less adherent. Although modest, our findings of less than 20% of participants having their CD4 count below 500 at the final visit are encouraging, and in line with previous findings in this population. In their cohort study, Diabaté et al. (2011), when comparing the immunologic response to ART between FSWs and the general population at a time when, in Benin, ART was only initiated in patients with CD4 < 200 cells/μl, observed a median increase in CD4 count of 103 cells/μl from a baseline of 134 cells/μl among FSWs over a one-year period [15]. This increase however, was somewhat lower compared to that of the general population (103 cells/μl versus 129 cells /μl; p = 0.085).

Using the ANRS-AC 11- HIV genotypic drug resistance interpretation's algorithms, [23] a prevalence of 12.2% for primary drug resistance prior to ART initiation was quite high compared to results from a previous study conducted in the same population [44]. In that study, Chamberland et al. found a prevalence rate of 3.9% for resistance mutations, with no major resistance to NRTIs, two resistances (K103N, G190A) to NNRTIs, and only one (F53Y) to protease inhibitors. Compared to our findings, in addition to K103N, and G190A, new resistant mutants included V179E, Y181C, Y181YF, Y181F, A98G, Y188L, P225H, and Y181S for NNRTIs, while those associated with resistance to NRTIs included M41L, D67N, T69D, 70R, M184V, T215F, K219Q, T215TS, and M184I. These findings point out both emergences of new circulating mutants, and/or arrival of new resistant mutants from surrounding countries where other ARV regimens including PIs are already in use. However, this high prevalence should be interpreted with caution since most participants with primary resistance were Beninese (only three Togolese). These could have initiated ART prior to inclusion in our study but hid the information during enrollment, as the services offered in our study could have been better than those provided in standard care. Luckily, this rate of primary drug resistance

observed at baseline in our study did not seem to be much alarming since none of participants diagnosed with resistance mutations, developed clinical resistance during the study, and even better, those who completed their final visit had all suppressed viral load and CD4 count above 500 cell/μl at their final visit. Two participants who developed drug resistance during follow-up had treatment failure (repeated viral load >1000).

## Study limitations

Two major caveats could have affected our results during this demonstration study project. The first is related to the high number of dropouts observed from the beginning to the end of the study reaching up to 40%. These dropouts were mainly due to the mobility of sex workers who either returned to their countries of origin or traveled to other cities to change work settings. In addition, we were not able to have access to data related to adherence, CD4 count, and viral load for those participants who went back to their countries of origin where they could also have accessed ART program and continued treatment. These losses to follow-up could have reduced the power of the study and have impact on our results since drop out subjects could be differentially related to treatment adherence and treatment response, causing either an overestimation or an underestimation of the association.

The second is related to the method of assessing adherence which used self-report. Indeed, self-reported adherence is subject to social desirability bias but also recall bias. These biases could overestimate adherence levels and affect the association between viral load and adherence to treatment. However, although not showing a clear trend of increasing viral load suppression with increasing adherence, especially in the upper adherence categories, the association we found between self-reported adherence and below quantification limit viral load is somewhat reassuring about the validity of self-reported adherence.

## Conclusion

Findings from this TasP demonstration project on FSWs indicate that immediate HIV treatment initiation following diagnosis is widely accepted, while feasibility could be affected by FSWs mobility and treatment adherence. Considering the sub-optimal follow up observed in this highly mobile population, a regional collaboration between FSW-friendly clinics is needed for sustained treatment implementation. We fell short of the UNAIDS objective of 90% viral suppression among those treated, underlining the need for better programs for enhancing treatment adherence, including structural interventions for reducing stigma and discrimination towards female sex workers and HIV-infected people.

## Key messages

Our results show that immediate treatment as prevention is quite accepted among female sex workers in Cotonou, Benin, and that suppressed viral load can be achieved in over 80% of female sex workers after one year, with good recovery in CD4 count and low emergence of drug resistance.

The study also highlights the need to consider the mobility of sex workers, their access to ARVs in their workplaces. It thus underlines the necessity of a regional collaboration between FSWs dedicated STI-clinics for effective implementation and effectiveness of TasP in FSWs in the sub-region.

## Supporting information

**S1 Appendix. Excel data to reproduce our results.**
(XLSX)

## Acknowledgments

We thank all participants in Benin, and Canada, the Benin research team, the staff from the IST clinic in Cotonou, and peer educators for their contribution; and all female sex workers who participated in the study. We also thank the laboratory investigators and study staff for their assistance with sample management and laboratory testing.

## Author Contributions

**Conceptualization:** Luc Béhanzin, Fernand A. Guédou, Michel Alary.

**Data curation:** Nassirou Geraldo, Dramane Kania, René Kpèmahouton Kêkê, Moussa Bachabi, Dissou Affolabi, Souleymane Diabaté, Flore Gangbo, Marcel Djimon Zannou.

**Formal analysis:** Mamadou Diallo, Ella Goma-Matsétsé.

**Writing – original draft:** Mamadou Diallo.

**Writing – review & editing:** Mamadou Diallo.

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
