## [Decision Letter · Decision Letter 0]

28 Jul 2019

PONE-D-19-15725

HIV treatment response among female sex workers participating in a treatment as prevention demonstration project in Cotonou, Benin

PLOS ONE

Dear Michel Alary,

Thank you for submitting your manuscript to PLOS ONE. After careful consideration, we feel that it has merit but does not fully meet PLOS ONE’s publication criteria as it currently stands. Therefore, we invite you to submit a revised version of the manuscript that addresses the points raised during the review process.

We would appreciate receiving your revised manuscript by Sep 11 2019 11:59PM. To enhance the reproducibility of your results, we recommend that if applicable you deposit your laboratory protocols in protocols.io, where a protocol can be assigned its own identifier (DOI) such that it can be cited independently in the future. For instructions see: http://journals.plos.org/plosone/s/submission-guidelines#loc-laboratory-protocols

We look forward to receiving your revised manuscript.

Kind regards,

Orna Mor

Academic Editor

PLOS ONE

Journal Requirements:

1. We note that you have indicated that data from this study are available upon request. PLOS only allows data to be available upon request if there are legal or ethical restrictions on sharing data publicly. For information on unacceptable data access restrictions, please see http://journals.plos.org/plosone/s/data-availability#loc-unacceptable-data-access-restrictions.

Additional Editor Comments (if provided):

The authors report on the results of a TasP demonstration study among a small cohort of 107 HIV-infected, treatment naïve, female sex workers in Cotonou, Benin. In general the study is clear and well written and interesting. However, the number of tables versus size of study is uncomparebly high. Table 1 could be removed and explained in the text; A graph of vl and cd4 versus visits could replace tables 3 and 4. Table 5 is unclearly presented. On the other hand, a table summarizing the prevlence and type of baseline DRM in the different class of drugs versus subtypes is missing. The outline of Table 6 is also unclear and if a new summary table is provided- than table 6 can be in appendix. Please also check if all references are correctly written and according to PLOS policy.

Reviewers' comments:

Reviewer's Responses to Questions

**Comments to the Author**

1. Is the manuscript technically sound, and do the data support the conclusions?

Reviewer #1: Partly

Reviewer #2: Yes

2. Has the statistical analysis been performed appropriately and rigorously? 

Reviewer #1: I Don't Know

Reviewer #2: Yes

3. Have the authors made all data underlying the findings in their manuscript fully available?

Reviewer #1: Yes

Reviewer #2: No

4. Is the manuscript presented in an intelligible fashion and written in standard English?

Reviewer #1: Yes

Reviewer #2: Yes

5. Review Comments to the Author

Reviewer #1: General comments:

The authors report on the results of a Treatment as Prevention demonstration study among a cohort of 107 HIV-infected, treatment naïve, female sex workers in Cotonou, Benin.

Their findings on acceptability, retention and viral load provide important and relevant information which should be taken into account when implementing TasP among key populations in Africa.

The general methods seem adequate. Statistical methods and analysis however should be verified. Some of the statistical tests used are not mentioned. The authors use generalized estimating equations to analyse time trends and the association between adherence and virological response however the results of the model are not clear and complete. It is also a pity that no attempt for identifying or adjusting for confounders have been made.

Specific comments:

Introduction

- P5, line 59: “an estimated 6.1 million people living with HIV at the end of 2016”

Please update the numbers and add a reference.

- P5, line 73: “(12,13).(14)”

Please correct this typo.

Methods

- P.6, line 90: “FSW not known to be HIV-positive on ART”

This expression is not clear, why the specification “on ART”?. Do you mean “FSW not known to be HIV-positive” ?

- P. 8, lines 130-132. Syphilis tests.

There may be some confusion when using the terms “rapid test” and “RPR”. Can you make it clear that the rapid is a treponemal, and the RPR is a nontreponemal test. One could also argue that one positive RPR test is not sufficient to detect active syphilis. According international guidelines there is a need of a fourfold change in titer, equivalent to a change of two dilutions to demonstrate an active infection.

Results

- P. 13, line 237: “13.2 ± 7.7”

Please specify that 7.7 is a standard deviation (I guess)

- P. 13, line 241: “the main reasons for not completing final visits”

The reason for not completing final visits is not the main issue I guess, but the reason for not being retained on ART, whether they come for a final visit or not.

- P. 15, line 277: “were infected or had a history of NG/CT”

Was this a history of an infection “ever”? It would make more sense to separate the biological results (STI at baseline) and the interview data “did you ever had an STI”, because they are two separate things.

- P .20, table 3 “Mean”

Please add “Mean CD4” for clarity in the title and the column head

- P. 20, table 3: p-values

The statistical methods (tests) for this table are not shown, please indicate which tests were used to obtain both type of p-values (p-trend and p-value), and explain in the methods.

- P. 23, table 4: p-values

Idem as previous comment.

- P. 24, lines 363-367: “the proportion … increased with increasing adherence (p=0.06 and 0.003)”

The results in the table do not confirm this statement: the proportion of visits where viral load was suppressed was 83.2% for the group with 90% adherence and 84.2% for the group 75-89% adherent.

- P. 24, table 5: results GEE

The table is quite difficult to read. The column with the head “Ratio” seems to present prevalences, not prevalence ratios. A prevalence is also a ratio in theory, but in this situation the term “ratio” adds to confusion.

- P. 24, table 5, p for trend (p=0.06 and 0.003)

Which test has led to these p’s? It is curious to see a p for trend of 0.06, when the PR are exactly the same in all adherence categories (with the exception of the reference category). Idem for the p for trend 0.03 for the outcome “undetectable”, where PR go from 3.2 for the group with the best adherence, vs 3.3 for the group with adherence 75% and 50%.

- P. 24, table 5: results GEE

The table only show the PR of the virological response per adherence level, the time factor is not presented. Have the authors considered to check potential confounders including age, behaviour factors such as condom use ? This may provide interesting alternative explanations for the relation adherence/virological outcome.

- P. 28-30, table 6

Please revise lay-out of the table, as the text in the lines is not positioned in a standardized way, which make the table difficult to read.

Discussion

- P. 31, line 440-441

Revise structure. The logical flow can be improved by replacing: “since half of them are not Beninese, and even when they were …”

- P. 32, line 453: “mobility”

Mobility is indeed be a serious barrier. The authors may discuss here the access of the FSW to ART services if they move to another place, another country.

- P. 32, line 466

Correct typo: “fact”

- P. 34, lines 509-512: “The results revealed … interventions”

This paragraph is repeated later in the discussion. It is a general conclusion and should be moved to the end of the paper.

- P. 35, lines 527-529

The authors should be cautious in their conclusions about the association between self-reported adherence and viral load, taking into account the results (see higher)

Reviewer #2: General comments

1. Although the study describes a TasP/PrEP demonstration project, limited data are provided about PrEP

2. The accession numbers of the nucleotide sequences generated in this project are not provided

3. The number of Tables should be reduced

Page 26, HIV strains and drug resistance (details in Table 6)

1 Authors provide subtype classification twice (i.e. CRF02_AG/CRF02_AG). This should be explained

2 The prevalence of resistance mutations should be reported

6. PLOS authors have the option to publish the peer review history of their article (what does this mean?). If published, this will include your full peer review and any attached files.

Reviewer #1: No

Reviewer #2: No

---

## [Author Response · Author response to Decision Letter 0]

27 Sep 2019

Dr Orna Mor, Ph.D. 

Academic Editor

PLoS ONE

RE: Response to reviewers - Manuscript ID: PONE-D-19-15725

Dear Editor,

We would like to sincerely thank the editor and the reviewers for the comprehensive evaluation of our manuscript entitled “HIV treatment response among female sex workers participating in a treatment as prevention demonstration project in Cotonou, Benin” that has been submitted for publication in PLOS ONE. 

We are excited the reviewers found our manuscript to be of significance to the field. Their pertinent comments and suggestions have definitely contributed to strengthen our paper, and as such, we have done our best to address all the editor’s and reviewers’ concerns. These changes are highlighted in the manuscript with the Word “Track changes” function. Since major revisions were brought and as requested in the letter from the academic editor, a “clean” version of the manuscript is also provided. As the revised manuscript also includes two figures, we have uploaded the figure files separately and have written the figure title and legends following the text that explains about the figure. 

Please find below the comments and questions (numbered and italicized) followed by our answers and details about the corrections that we performed on the original manuscript. Please note that when we refer to line numbers, those refer to line numbers as read in the clean version of our revised paper without tracked changes.

The revised manuscript is now ready for further consideration by PLoS ONE. We sincerely hope that it will now be suitable for publication in your very interesting journal. 

Best Regards,

Mamadou Diallo and Michel Alary (on the behalf of all coauthors)

Note from the Editor:

1) We note that you have indicated that data from this study are available upon request.

Response: Indeed, as we mentioned while submitting the manuscript, data are available and can be made accessible. We have now provided a minimal anonymized data set necessary to replicate our study findings. This file uploaded as a supporting Information file can be accessed in the PloS ONE website.

Additional Editor Comments (if provided):

The authors report on the results of a TasP demonstration study among a small cohort of 107 HIV-infected, treatment naïve, female sex workers in Cotonou, Benin. In general, the study is clear and well written and interesting. However, the number of tables versus size of study is uncomparebly high.

2) Table 1 could be removed and explained in the text

Response: We have removed the table 1 and explained the corresponding data in the text (page 12-13: line 230-233). 

3) A graph of cd4 and vl versus visits could replace tables 3 and 4. 

Response: We have replaced tables 3 and 4 by Figures 1 and 2 (2 figures needed to provide enough details) that we have uploaded as separate files. Figure titles and legends are provided in the manuscript itself (see page 19, lines 282-289 for changes in CD4, and page20-21: lines 313-318 for viral response). We have also slightly modified the text describing these 2 figures to provide some of the actual numbers that were in the table and that are somewhat less clear in a figure format (see page 18-19, lines 269-276 for changes in CD4 and page 20, lines 300-308 for viral load).

4) Table 5 is unclearly presented

Response: We have entirely reformatted table 5, which is now table 2 (see page 21). 

5) On the other hand, a table summarizing the prevalence and type of baseline DRM in the different class of drugs versus subtypes is missing. 

Response: The information on baseline DRM was already present in former Table 6; indeed, we agree it was badly formatted. We have also entirely reformatted this table that is now Table 3 (see page 24-25). In addition, we now provide data on baseline prevalence of DRM in the text (page 23, lines 350-367).

6) The outline of Table 6 is also unclear and if a new summary table is provided- then table 6 can be in appendix. 

Response: See our response to the previous comment. 

7) Please also check if all references are correctly written and according to PLOS policy.

Response: We have checked the references and adjusted them to PLOS policy where necessary.

Reviewer #1: General comments: 

8) The authors report on the results of a Treatment as Prevention demonstration study among a cohort of 107 HIV-infected, treatment naïve, female sex workers in Cotonou, Benin. Their findings on acceptability, retention and viral load provide important and relevant information which should be taken into account when implementing TasP among key populations in Africa. The general methods seem adequate. Statistical methods and analysis however should be verified. Some of the statistical tests used are not mentioned. The authors use generalized estimating equations to analyse time trends and the association between adherence and virological response however the results of the model are not clear and complete. It is also a pity that no attempt for identifying or adjusting for confounders have been made.

 Response: We thank the reviewer for addressing this aspect of the methodology of data analysis, and his willing to have more information in the methodology. We hereby provide detail information on the methodology and reasons we used for the analysis. Indeed, since we had repeated data, the use of generalized estimating equations with the log –link and the type3 test of Wald seemed to be more appropriate for the analysis of time trend changes as well as the association between viral load and self-reported adherence. While analyzing the association between viral load and self-reported adherence, the final model controlled for time duration since ART initiation. We also initially included potential confounders such as age, country of origin, duration in sex work, having a regular partner (or not), and number of clients (but not reported condom use as it was reported to be extremely high: see Table) in the multivariate model, however, all these variables did not confound the association between self-reported adherence and viral load. Therefore, we did not keep them in the final model in order to maximize statistical power. We have made some changes in the section on statistical methodology to better describe our approach (see page 11, lines 197-202) as well as in the results section when reporting the associations between self-reported adherence and viral load (see page 21, lines 324-332).

Specific comments:

Introduction

9) - P5, line 59: “an estimated 6.1 million people living with HIV at the end of 2016”

Please update the numbers and add a reference.

Response: We have updated both the numbers and the reference. (Page 5: line 58-60). 

10) - P5, line 73: “(12,13).(14)”

Please correct this typo.

Response: Corrected (page 5: line 73).

11) Methods

- P.4, line 93: “FSW not known to be HIV-positive on ART”

This expression is not clear, why the specification “on ART”? Do you mean “FSW not known to be HIV-positive”?

Response: We thank the reviewer for pointing out this ambiguity. Indeed, this expression was misplaced. It’s only after the confirmation of the screening test at the Dispensaire IST, that we restrained recruitment to HIV-positive FSWs who were not known to be on ART. Being ART naïve were confirmed following responses to questions, and confirmation the clinic records as the vast majority of HIV-positive FSWs in Cotonou who are on ART receive their treatment at Dispensaire IST). In the manuscript, the deletion at the inappropriate location of the manuscript is on page 6: line 90-91 and the addition is on page 6, lines 93-95. 

IST (in order to confirm that participants were ART-naïve)

12) - P. 8, lines 130-132. Syphilis tests.

There may be some confusion when using the terms “rapid test” and “RPR”. Can you make it clear that the rapid is a treponemal, and the RPR is a nontreponemal test. One could also argue that one positive RPR test is not sufficient to detect active syphilis. According international guidelines there is a need of a fourfold change in titer, equivalent to a change of two dilutions to demonstrate an active infection.

Response: We have now specified that the rapid test was treponemal and that the RPR was non-treponemal (page 8: Line 130-133). Note also than in cross-sectional studies (and all data presented in table 1 are from baseline and thus cross-sectional), it is usual to apply the definition we used for active syphilis. We don not report follow-up data on this issue, but please note that, in addition to not having a single case of active syphilis at baseline, there was no new case of syphilis during follow-up. 

Results

13) - P. 13, line 237: “13.2 ± 7.7”

Please specify that 7.7 is a standard deviation (I guess)

Response: This is now specified (page 12: line 230). 

14) - P. 13, line 241: “the main reasons for not completing final visits”

The reason for not completing final visits is not the main issue I guess, but the reason for not being retained on ART, whether they come for a final visit or not.

Response: In fact, we cannot know if the women who did not complete follow-up in the study were retained on ART or not. It depends if they accessed ARV clinics after leaving the study (most of them were not in Cotonou anymore and many were in other countries). Therefore, the reasons given in the manuscript are for withdrawing from the study. The wording of this section has also been changed because of the deletion of original table 1 and the addition of new text following a request from the editor (see page 9: 230-236).

15) - P. 15, line 277: “were infected or had a history of NG/CT”

Was this a history of an infection “ever”? It would make more sense to separate the biological results (STI at baseline) and the interview data “did you ever had an STI”, because they are two separate things.

Response: We thank the reviewer for pointing out this mistake. The prevalence we provided was in fact for the actual biological results of the nucleic acid amplification tests for NG and CT and did not include past history collected by questionnaire. This is now corrected (see page 13, lines 249-252). 

16) - P .20, table 3 “Mean”

Please add “Mean CD4” for clarity in the title and the column head

Response: As per the editor’s request, we have replaced table 3 by figure 1, but mean CD4 count is now specified in this figure’s title and legend (page 19: Line 282-292).

17) - P. 20, table 3: p-values

The statistical methods (tests) for this table are not shown, please indicate which tests were used to obtain both type of p-values (p-trend and p-value), and explain in the methods. 

Response: We described the analysis methodology in the method section. For all the analysis, we used generalized estimating equations with the log–link and the type3 test of Wald (see page 11: Lines 197-202), (see also our response to comment 8 above). Note that table 3 has now been replaced by figure 1.

18) - P. 23, table 4: p-values

Idem as previous comment.

Response: Same response as to the previous comment. Note also that table 4 has now been replaced by figure 2. (see page 11: Lines 197-202).

19) - P. 24, lines 363-367: “the proportion … increased with increasing adherence (p=0.06 and 0.003)”. The results in the table do not confirm this statement: the proportion of visits where viral load was suppressed was 83.2% for the group with 90% adherence and 84.2% for the group 75-89% adherent.

Response: We used 4 adherence categories in this analysis and evaluated a test for trend over the 4 categories and these are the p-values we obtained (see our response to comment 8 above). This can happen even if the various categories yield similar prevalence ratios when compared to the reference category as long as the association is strong enough. However, we agree with the reviewer in the fact that it would be important to qualify this statement by evoking the stability of the prevalence ratios as long as the adherence level is >50%. We have thus added a sentence in this regard in the result section (see page 21, lines 330-332). 

20) - P. 24, table 5: results GEE

The table is quite difficult to read. The column with the head “Ratio” seems to present prevalences, not prevalence ratios. A prevalence is also a ratio in theory, but in this situation the term “ratio” adds to confusion.

Response: As per the editor’s request, we have completely reformatted this table, that is now table 2, and we have changed the column header as suggested by the reviewer. (see page 21-22: line 334-340).

21) - P. 24, table 5, p for trend (p=0.06 and 0.003) 

Which test has led to these p’s? It is curious to see a p for trend of 0.06, when the PR are exactly the same in all adherence categories (with the exception of the reference category). Idem for the p for trend 0.03 for the outcome “undetectable”, where PR go from 3.2 for the group with the best adherence, vs 3.3 for the group with adherence 75% and 50% (page 15: line 334).

Response: See our response to comments 8, 17 and 19 above. 

22) - P. 24, table 5: results GEE

The table only show the PR of the virological response per adherence level, the time factor is not presented. Have the authors considered to check potential confounders including age, behaviour factors such as condom use? This may provide interesting alternative explanations for the relation adherence/virological outcome.

Response: As time since treatment initiation was treated as a confounding factor for the association between adherence levels and viral load and that we were mainly interested by this latter association, we do not present prevalence ratios for this time variable to keep the table as simple as possible for the readers. Concerning the other potential confounders, see our response to comment 8 above.

23) - P. 28-30, table 6

Please revise lay-out of the table, as the text in the lines is not positioned in a standardized way, which make the table difficult to read.

Response: This table has now been entirely reformatted and its number as changed to table 3 (see page 24-26).

Discussion

24) - P. 31, line 440-441

Revise structure. The logical flow can be improved by replacing: “since half of them are not Beninese, and even when they were …”

Response: We have reviewed and rephrased the sentence as suggested (page28: line 418-423).

25) - P. 32, line 453: “mobility”

Mobility is indeed be a serious barrier. The authors may discuss here the access of the FSW to ART services if they move to another place, another country.

Response: We added some discussion on this issue (see page 29, lines 433– 443). 

26) - P. 32, line 466

Correct typo: “fact”

Response: Corrected (page 29: line 433)

27) - P. 34, lines 509-512: “The results revealed … interventions”

This paragraph is repeated later in the discussion. It is a general conclusion and should be moved to the end of the paper.

Response: We removed this paragraph as suggested by the reviewer (see page 32, lines 499-502). At the same time, since this sub-section now only present limitations, we changed the name of the sub-section “Study strengths and caveats” to “Study limitations”.

28) - P. 35, lines 527-529

The authors should be cautious in their conclusions about the association between self-reported adherence and viral load, taking into account the results (see higher)

Response: We have made some changes in order to be more cautious with this conclusion (see page 32, line 506-510).

Reviewer #2: General comments

29) Although the study describes a TasP/PrEP demonstration project, limited data are provided about PrEP

Response: We thank the reviewer for his interest in the PrEP data. Although our project focused on TasP and PrEP, we only report data on TasP in this paper. Indeed, we have already published a paper on the overall resust of our TasP/PrEP study in the Journal of the International AIDS Society (JIAS) (see 20 of the present study or DOI: 10.1002/jia2.25208). The title of this paper is “Early antiretroviral therapy and daily pre-exposure prophylaxis for HIV prevention among female sex workers in Cotonou, Benin: a prospective observational demonstration study. However, the results presented in the present paper include some additional TasP subjects and present analyses that are not included in our JIAS paper.

30) The accession numbers of the nucleotide sequences generated in this project are not provided

Response: We perfectly agree with the reviewer that the nucleotide sequences generated in this project should be provided. Unfortunately, we did not think about recording these sequences in Genbank. However, if requested, we can provide s the sequences for verification. 

31) The number of Tables should be reduced

Response: This has been done: we went from 6 tables to 3 tables and 2 figures following the editor’s comments. 

32) Page 26, HIV strains and drug resistance (details in Table 6)

Authors provide subtype classification twice (i.e. CRF02_AG/CRF02_AG). This should be explained

Response: As to our understanding of the question, the reviewer asks us to explain why we reported the same subtype twice (i.e. CRF02_AG/CRF02_AG). This is a standard terminology for presenting the results of genotyping for drug resistance. Indeed, sequencing the HIV-1 pol and env genes is routinely used to identify mutations associated with resistance to reverse transcriptase (RT) or protease (PR) inhibitors in HIV-1 circulating recombinant forms (CRFs) including CFR02_AG and CRF01_AE. Samples were genotyped at both env and pol regions. When the results yielded identical sequences in the two regions, we reported the same sequence twice (ex.: CRF02_AG/CRF02_AG), otherwise we reported differently (i.e. CRF11-cpx/CRF02_AG). This appears in the revised table 3 (i.e. G/CRF02_AG and CRF11-cpx/CRF02_AG) and shows the different strains in our study.

The prevalence of resistance mutations should be reported

 Response: We now report the prevalence of resistance mutations in the text of the manuscript (see page 22-23, lines 345-353).

---

## [Decision Letter · Decision Letter 1]

25 Oct 2019

PONE-D-19-15725R1

HIV treatment response among female sex workers participating in a treatment as prevention demonstration project in Cotonou, Benin

PLOS ONE

Dear Dr. Alary,

Thank you for submitting your manuscript to PLOS ONE. After careful consideration, we feel that it has merit but does not fully meet PLOS ONE’s publication criteria as it currently stands. Therefore, we invite you to submit a revised version of the manuscript that addresses the points raised during the review process.

We would appreciate receiving your revised manuscript by Dec 09 2019 11:59PM. To enhance the reproducibility of your results, we recommend that if applicable you deposit your laboratory protocols in protocols.io, where a protocol can be assigned its own identifier (DOI) such that it can be cited independently in the future. For instructions see: http://journals.plos.org/plosone/s/submission-guidelines#loc-laboratory-protocols

We look forward to receiving your revised manuscript.

Kind regards,

Orna Mor

Academic Editor

PLOS ONE

Additional Editor Comments (if provided):

Thanks you for submitting the revised manuscript. The previous corrections requested by the reviewers were met. However, there are still few minor remarks on the revised version. I do not find that the correction requested by reviewer 1 is necessary as indeed as stated and discussed t is possible that some of the FSW were on ART previous the study. Therefore I suggest to leave the sentence "HIV positive FSW not known to be on ART"

Regarding the remark of reviewer 2- I agree, if it is possible.

The followings are few remarks that if corrected, could better clear this paper:

Line 139: Neisseria gonorrhoeae (NG) and Chlamydia trachomatis (CT). In Table 1 : N. gonorrhoeae or C. trachomatis. Please choose either NG/CT or N. gonorrhoeae - C. trachomatis throughout the text.

Page 9 148-151: nuclisens can officially assess HIV-1 only (not HIV-2). Only one publication (ref 21) assessed its performance on HIV-2 subtype A samples only and claims detection limit of 200 c/ml. This is not what is stated here. I suggest to remove HIV-2 from this sentence.

Also the limit of quantitation for HIV -1 has changed during the years. Please provide the version number of the VL kit used in this study and give the updated LOQ for the version used herein (as far as I recall it was never 40c/ml).

LINE 151: Replace HIV by HIV-1 (to exclude HIV-2)

Line 179: vl <40 cannot considered undetectable. If 40 is the quantification limit It could be considered as below quantification limit. If <40 is the LOQ for the assay used herein, please amend the term “undetectable” ( in cases of VL <40 C/ML) to “below quantification limit” all through this paper.

Line 282: Fi 1 change to Figure 1

Line 296: term “suppressed viral load” and “undetectable viral load” should be explained in the text and not only in the figure legend, especially as <1000 is not regularly considered “suppressed…”. Please discuss to these numbers in the discussion section as today VL >200 (SOMETIMES >50) is considered virological failure.

Table 1 : need restructuring. Numbers are not in lanes and some do not fit the space

Table 3: needed reformatting and detailing. What does NA mean? Why some spaces are empty? The list of ART for each sample is the drugs which could be taken and are no affected by the identified DRM? Why to include drugs that are not in use any more? The subtype is given for both RT and PR? Please provide detailed information so that the reader will easily understand.

Reviewers' comments:

Reviewer's Responses to Questions

**Comments to the Author**

1. If the authors have adequately addressed your comments raised in a previous round of review and you feel that this manuscript is now acceptable for publication, you may indicate that here to bypass the “Comments to the Author” section, enter your conflict of interest statement in the “Confidential to Editor” section, and submit your "Accept" recommendation.

Reviewer #1: All comments have been addressed

Reviewer #3: All comments have been addressed

2. Is the manuscript technically sound, and do the data support the conclusions?

Reviewer #1: Yes

Reviewer #3: Yes

3. Has the statistical analysis been performed appropriately and rigorously? 

Reviewer #1: Yes

Reviewer #3: Yes

4. Have the authors made all data underlying the findings in their manuscript fully available?

Reviewer #1: Yes

Reviewer #3: Yes

5. Is the manuscript presented in an intelligible fashion and written in standard English?

Reviewer #1: Yes

Reviewer #3: Yes

6. Review Comments to the Author

Reviewer #1: Just one small correction to make: revised clean text, line 93 "HIV positive FSW not known to be on ART" is confusing, should be "HIV positive FSW not on ART"

Reviewer #3: This is a strong paper that I enjoyed reading. The one remaining issue that I would strongly recommend the authors to consider is to present a comparison of baseline characteristics of participants who did and not complete the follow-up to give readers a better sense of the types of participants who dropped out of the study.

7. PLOS authors have the option to publish the peer review history of their article (what does this mean?). If published, this will include your full peer review and any attached files.

Reviewer #1: No

Reviewer #3: Yes: Timothy P Johnson

---

## [Author Response · Author response to Decision Letter 1]

9 Dec 2019

Note from the Editor:

1) Page 8 line 139: Neisseria gonorrhoeae (NG) and Chlamydia trachomatis (CT). In Table 1: N. gonorrhoeae or C. trachomatis. Please choose either NG/CT or N. gonorrhoeae - C. trachomatis throughout the text.

Response: We have chosen to use N. gonorrhoeae and C. trachomatis throughout the text (page 8, line 140-141; page 14 line 257; table 1); except for the name of the Probetec assay used to test for these infections, as NG/CT is part of the brand name of the assay. 

2) Page 9 148-151: nuclisens can officially assess HIV-1 only (not HIV-2). Only one publication (ref 21) assessed its performance on HIV-2 subtype A samples only and claims detection limit of 200 c/ml. This is not what is stated here. I suggest to remove HIV-2 from this sentence.

Response: Indeed, Biomerieux-diagnostics mentioned that NucliSens® HIV-1 QT assay is generally insensitive for detection of HIV-2 RNA. In addition, samples from individuals infected with HIV-2 may exhibit cross-reactivity in this assay. We therefore, have removed HIV-2 from this sentence, as suggested (page 9: line 151). Please also note that, as already stated in the manuscript, subjects with evidence of HIV-2 infection were excluded from the study.

3) Also the limit of quantitation for HIV-1 has changed during the years. Please provide the version number of the VL kit used in this study and give the updated LOQ for the version used herein (as far as I recall it was never 40c/ml).

Response: We used the NucliSens EasyQ® HIV-1 v2.0 kit. 48 tests - Ref: 28 5033. The kit is of high-level sensitivity with validated input volumes plasma, can detect the majority of HIV-1 subtypes including: A, B, C, D, F, G, H, J, CRF01_AE and CRF02_AG with a viral load measuring range between: 10-10 000 000 copies/ml. 

This kit provides detection limits according to volume of plasma used:

Flexible Input Limit of Detection (LoD)

1 ml plasma 25 cps/ml

0.5ml plasma 50 cps/ml

0.1 ml plasma 292 cps/ml

Dry Blood Spot (0.1 ml) 802 cps/ml

We used 0.5 ml plasma, therefore the lowest viral load we could amplify was 50 copies/ml (page 9: line 151). 

4) Page 9 line 151: Replace HIV by HIV-1 (to exclude HIV-2)

Response: We have replaced HIV by HIV-1 (page 9, line 152). 

5) Page 10 line 179: vl <40 cannot considered undetectable. If 40 is the quantification limit it could be considered as below quantification limit. If <40 is the LOQ for the assay used herein, please amend the term “undetectable” (in cases of VL <40 C/ML) to “below quantification limit” all through this paper.

Response: Thank you very much for underlining this ambiguous misuse of “undetectable” instead of “below quantification limit”, “undetectable” being a term too often used in viral quantification. Since we used the NucliSens EasyQ® HIV-1 v2.0 kit. 48 tests - Ref: 28 5033 and 0.5 ml plasma, the lowest viral load we could amplify was 50 copies/ml. We therefore have amended the term “undetectable” (in all cases of VL <40 C/mL) to “below quantification limit” all through the manuscript, and the figure 2. Please, see (page 3: line 43-45; page 9: line 151; page 10 line 178-185; page 11 line 201-202; page 18 line 304-307; page 19 line 313-327; page 20 line 338-340, table 2; page 26 line 397).

6) Page 18 line 282: Fi 1 change to Figure 1

Response: We have changed Fi 1 to Figure 1 (page 18: line 290).

7) Page 19 line 296: term “suppressed viral load” and “undetectable viral load” should be explained in the text and not only in the figure legend, especially as <1000 is not regularly considered “suppressed…” Please discuss to these numbers in the discussion section as today VL >200 (SOMETIMES >50) is considered virological failure.

Response: The term viral suppression or “suppressed viral load”, when viral load is <1000 copies/mL was used to evaluate treatment success by the World Health Organisation (WHO) and the Joint United Nations Programme on HIV/AIDS (UNAIDS). According to the 2016 WHO Consolidated guidelines on the use of antiretroviral drugs for treating and preventing HIV infection, a viral load <1000 copies/mL defines treatment success. The term “undetectable” viral load was defined as viral load below detection limits of the assay used for viral load quantification in the study (page 10 line 180-184, page 26 line 400-402). For the latter, we however now use “below quantification limit” instead of “undetectable” throughout the manuscript.

8) Table 1: Needs restructuring. Numbers are not in lanes and some do not fit the space

Response: We have restructured Table 1 so that numbers are in lanes and fit the space (page 14-17).

9) Table 3: needed reformatting and detailing. What does NA mean? Why some spaces are empty? The list of ART for each sample is the drugs which could be taken and are no affected by the identified DRM? Why to include drugs that are not in use any more? The subtype is given for both RT and PR? Please provide detailed information so that the reader will easily understand.

Response table 3: NA meant not applicable as there was no resistance mutation to this class of ARVs; some spaces were empty for the same reason. We have now decided to leave all these spaces empty when no resistance mutation was present to a given ARV class. The list of specific drugs is quite useless at the end and we have suppressed this column entirely in order to make the table easier to read. We have provided the sequences corresponding to the protease and the reverse transcriptase regions (PR/RT), and this is now better explained (table page 23-24: table 3).

---

## [Editor Report · Decision Letter 2]

16 Dec 2019

HIV treatment response among female sex workers participating in a treatment as prevention demonstration project in Cotonou, Benin

PONE-D-19-15725R2

Dear Dr. Alary,

We are pleased to inform you that your manuscript has been judged scientifically suitable for publication and will be formally accepted for publication once it complies with all outstanding technical requirements.

With kind regards,

Orna Mor

Academic Editor

PLOS ONE

Additional Editor Comments (optional):

Thank you for addressing all issue and correcting the paper. I have enjoyed reading it and look forward to seeing it on pubmed.

I still find Table 3 not lined up- but I guess you can improve it when the paper is processed for publication.
---

## [Editor Report · Acceptance letter]

8 Jan 2020

PONE-D-19-15725R2 

HIV treatment response among female sex workers participating in a treatment as prevention demonstration project in Cotonou, Benin 

Dear Dr. Alary:

I am pleased to inform you that your manuscript has been deemed suitable for publication in PLOS ONE. Congratulations! Your manuscript is now with our production department. 

With kind regards,

on behalf of

Dr. Orna Mor 

Academic Editor

PLOS ONE